# LOSSLESS SINGLE IMAGE SUPER RESOLUTION FROM LOW-QUALITY JPG IMAGES

## ABSTRACT

Super Resolution (SR) is a fundamental and important low-level computer vision (CV) task. Different from traditional SR models, this study concentrates on a specific but realistic SR issue: How can we obtain satisfied SR results from compressed JPG (C-JPG) image, which widely exists on the Internet. In general, C-JPG can release storage space while keeping considerable quality in visual. However, further image processing operations, e.g., SR, will suffer from enlarging inner artificial details and result in unacceptable outputs. To address this problem, we propose a novel SR structure with two specifically designed components, as well as a cycle loss. In short, there are mainly three contributions to this paper. First, our research can generate high-qualified SR images for prevalent C-JPG images. Second, we propose a functional sub-model to recover information for C-JPG images, instead of the perspective of noise elimination in traditional SR approaches. Third, we further integrate cycle loss into SR solver to build a hybrid loss function for better SR generation. Experiments show that our approach achieves outstanding performance among state-of-the-art methods.

## 1 INTRODUCTION

With the marvelous achievement of deep learning (DL) in computer vision (CV), Super Resolution (SR) attracts much attention for its crucial value as the basis of many high-level CV tasks (Chen et al., 2017; He et al., 2016). Deep learning Super Resolution (DL-SR) algorithms (Kim et al., 2016; Lim et al., 2017; Haris et al., 2018; Zhang et al., 2018c;b) strive for finding the complex nonlinear mapping between low resolution (LR) images and their high resolution (HR) counterparts. However, the learned model only reflects the inverse of down-scaled mapping, which is used to obtain LR images from their HR fathers. In other words, if there are some spots/stains in LR inputs, the SR model will treat them as inherent elements, and the corresponding SR outputs will enlarge these undesirable details. In reality, on the Internet, JPG compression is probably the most commonly used pattern for storage space reduction. That is to say, the LR image will be further processed into a compressed JPG (C-JPG) image. The quality of C-JPG will greatly drop, and the compression may yield unpleasant artifacts, for example, the presence of obvious partition lines, which vastly deteriorates the overall visual feeling. Hence, directly solving high-level CV tasks with these C-JPG images will lead to poor performance. In this paper, we propose a lossless SR model to obtain images with satisfying quality from the low-quality inputs (C-JPG). The deterioration in C-JPG makes the SR processing a huge challenge.

In this paper, we focus on the more realistic C-JPG SR problem. Many SR methods regarding to the real-world condition images have been already developed, such as Zhang et al. (2018a); Yuan et al. (2018). Among them, some models regard the noise as a kernel estimating problem which can be solved by addictive Gaussian noises. However, the distribution of most real images are inconsistent with the hypothetical Gaussian distribution. Taking C-JPG images as a example, the image compression operation is related to decreasing information from original image instead of adding specific noises. Other models learn the related information from irrelevant LR-HR images to obtain similar representations by unsupervised strategy. All of them cannot solve the problem well.

In general, most LR images are produced through performing traditional interpolation method (mostly bicubic) on their HR fathers. The SR training process should recover this down-scaled mapping in a reverse manner. Referring to our C-JPG SR issue, when searching images from Google, a

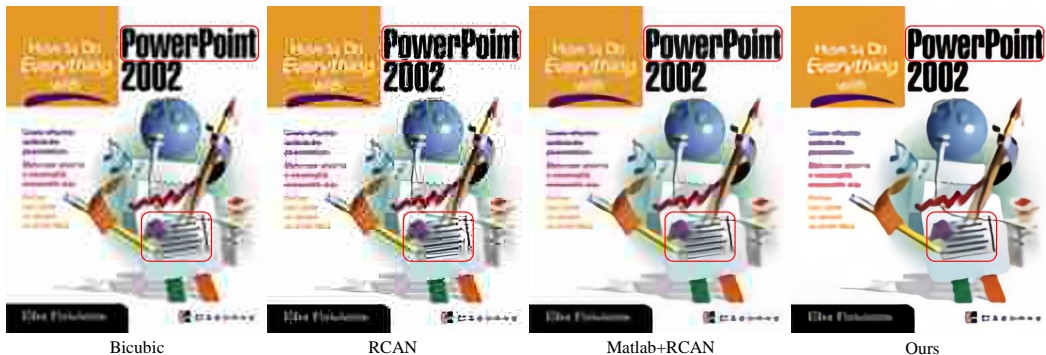

Figure 1: SR generations of four different settings: Bicubic, RCAN, RCAN with denosing input by Matlab, and our model. We marked some obvious details in red rectangles to indicate the advantage of our method.

lot of unpleasant details are displayed, especially in the edges of objects. However, the low quality of image makes former SR methods fail to generate applicable images. As shown in Fig. 1, it is shown that the SR generations of traditional bicubic interpolation, leading SR algorithm RCAN, and RCAN with pre-denoising input all demonstrate poor quality with the low quality C-JPG inputs. Damaged grids are apparently enlarged by the approaches designed for traditional non-JPG datasets. More specialized analysis can be found in the research of Köhler et al. (2017).

Note that the image pairs with fixed down-scaled kernel have been successfully learnt by SR models, such as SRGAN (Ledig et al., 2017), EDSR (Lim et al., 2017), and RDN (Zhang et al., 2018c). In this study, we deliberately build a more complicated dataset by adding JPG format LR images to the training data. To be specific, we have three kinds of training inputs: C-JPG LR, LR, and HR images. The whole training process includes two separate functional components: missing detail recuperative part (JPG recovering stage) and SR mapping learning part (SR generating stage). In order to remove ring, checkerboard effects, as well as other noise, the former half sub-model is trained with pre-processed C-JPG LR images as inputs, and original LR ones as the supervised information. The function of this stage is to recover LR image from its compression counterpart. Hence, the outputs ($LR(C - JPG)$) of the first part are greatly improved and free of partition lines phenomenon. Based on these improved LR images, the latter sub-model continues to learn the mapping between ($LR(C-JPG)$) and $HR$. Therefore, an integrated pipeline for SR representation between C-JPG and HR images is achieved through the jointly two sub-models.

In short, there are mainly three contributions in this study:

- Our research can be regarded as an universal SR method that generates SR images from C-JPG inputs, which is empirically proved to be more difficult than SR with non-JPG inputs.
- We regard this specific SR task as a recovering information process for the inputs, compared with the former denoising assumption including down-sampling and degradation parts. In this viewpoint, a recovering model is firstly introduced to generate satisfied intermediates from C-JPG inputs.
- We further propose an integrated SR model training pipeline with two-level data, i.e., C-JPG LR and LR images, as well as a new integrated loss function. The experimental results demonstrate our method can surpass traditional SR models.

## 2 RELATED WORK

### 2.1 IMAGE SUPER-RESOLUTION

Single image super-resolution (SISR) is a widely concerned task. A bundle of researches is developed to resolve this challenge, and great achievements have been achieved. At first, researchers obtain SR generations with an estimated pixel-value interpolation function. For instance, Bicubic regards pixel spreading in a limited patch as a specified linear parametric function. However, these

methods are prone to generate fuzzy SR results. Later, depending on external and internal dataset, two different classes of SR methods (external and internal based models) are developed. In detail, external models learn the mapping upon many image pairs with various learning algorithms, such as nearest neighbor (Freeman et al., 2002), kernel ridge regression (Kim & Kwon, 2010), sparse learning (Yang et al., 2010; 2012), cluster learning (Timofte et al., 2013; Yang & Yang, 2013), manifold learning (Chang et al., 2004), and neural networks (Dong et al., 2014). Internal models leverage similar patches within the image or across scales, and the approaches mainly focus on obtaining more self-similar patches, such as Ebrahimi & Vrscay (2007); Freedman & Fattal (2011); Irani (2009); Michaeli & Irani (2013); Yang et al. (2013); Huang et al. (2015).

Recently, deep learning method, which has achieved excellent performance in many high-level CV tasks, is introduced to SR. DL-SR focuses on learning the relationship between LR image and its corresponding HR one. To our knowledge, the most cutting-edge SR algorithms DL-SR, thanks to the powerful learning ability of deep learning. In detail, Dong et al. (2014) firstly propose end-to-end DL-SR network, SRCNN. Though there are only three convolutional layers in SRCNN, it greatly surpasses the former traditional methods in peak-signal-to-nosie-ratio (PSNR). Ever since SRCNN, a number of novel models arise for better generation ability, such as FSRCNN (Dong et al., 2016), VDSR (Kim et al., 2016), SRGAN (Ledig et al., 2017), EDSR (Lim et al., 2017), RDN (Zhang et al., 2018c), DBPN (Haris et al., 2018), ESRGAN (Wang et al., 2018), and RCAN (Zhang et al., 2018b).

In DL-SR, the goal is developed into two different aspects, i.e., higher accuracy factor (high PSNR and SSIM values) and more photo-realistic (better feature similarity) in visual sense. Based on this consideration, various structures are designed to extract more crucial features from the LR image. For example, VDSR (Kim et al., 2016) introduces the global residual learning and trains a multi-scale model through sharing parameters across different scales. Inspired by the stunning performance of ResNet (He et al., 2016), the SRGAN (Ledig et al., 2017) replaces most basic convolutional blocks with residual ones. Moreover, it combines MSE loss, GANs loss (Goodfellow et al., 2014) loss, perceptual loss (Johnson et al., 2016), and global loss for photo-realistic SR generations. Experiments show the great power in generating photo-realistic images. Based on the generator of SRGAN, EDSR (Lim et al., 2017) removes all batch normalization (Ioffe & Szegedy, 2015) to reduce the computation complexity, and replaces MSE loss with L1 norm loss. Moreover, the model increases the channel number to 256 in each layer and achieves state-of-the-art results in NTIRE2017 (Timofte et al., 2017). However, there are over 40 million parameters in EDSR. Benefit from the joint of densely architecture, such as Densely Network (Huang et al., 2017), RDN (Zhang et al., 2018c) surpasses EDSR in accuracy. RCAN (Zhang et al., 2018b) upgrades the densely block to residual in residual (RIR) block and combines with channel attention mechanism. To our knowledge, RCAN is the leading method in accuracy pursuing SISR. Meanwhile, some researchers aim to build a lightweight and accurate model, such as CARN (Ahn et al., 2018), s-LWSR (Li et al., 2019), and FALSR (Chu et al., 2019). These methods try to reduce parameters and operations while keeping the decent performance. CARN leverages the cascading mechanism upon a residual block for obtaining better feature representation from multi-level layers. At the same time, to decrease the model size, some residual blocks are replaced by group convolution parts, which is similar to depth-wise convolution in MobileNets (Howard et al., 2017). The s-LWSR tries to introduce a more flexible SR model with an additional information pool. Inspired by the NAS (Zoph & Le, 2016), FALSR automatically searches a desirably lightweight and accurate SR model without human-designing.

## 2.2 KERNEL UNKNOWN IMAGE SUPER-RESOLUTION

All methods mentioned above try to solve the SR problem on algorithmic LR-HR image pairs. However, the scale-kernel and degradation function are usually undefined in real-world, which results in the SR image accompanied by amplified noise.

To overcome this weakness, ZSSR (Shocher et al., 2018) proposes an unsupervised Zero-shot SR method. Firstly, the method generates many derived HR and LR images from the input and builds a simple CNN network to learn the mapping between pre-proposed LR image and its HR counterpart. So far as we know, ZSSR greatly surpasses other supervised SR models on the non-ideal condition mentioned above. Learning from CycleGAN (Zhu et al., 2017), Yuan et al. (2018) propose a cycle-in-cycle structure called CincGAN to address the blind SR issue. Inputs with noise are firstly processed to generate intermediate LR images with less noise. Then, these LR images are jointly restored and scaled up with the help of an extra pre-trained SR model. Besides, Shi et al. (2019)

proposes the multi-gram losses in UMGSR to generate perceptual satisfying images. For supervised method, Xu et al. (2019) propose a new pipeline to learn to generate SR images from the generated realistic data of raw image.

## 3 METHOD

**Challenge Formulation** In general, SISR issue can be formulated as: $y = \mathcal{F}(x) + z$, where $y$ and $x$ represent HR and LR images respectively. Here, $\mathcal{F}$ is the SR mapping in the hypothesis set space (e.g., neural networks), and $z$ denotes the additional irrelevant information, such as noises and blurred details. Normally, most SISR models are trained on the standard dataset, where the input LR is directly downsampled from HR by chosen method (e.g., Bicubic). As a result, we assume that $z$ equals to zero in the dataset. In this paper, we investigate a specific situation where the LR is low-quality C-JPG image. This situation commonly exists on the Internet, due to reducing storage or protecting copyright. Since LR inputs are firstly deteriorated to low-quality images, we redefine the above SISR formulation as: $y = \mathcal{F}(x + w)$. Here, $w$ refers to the missing information due to the compression.

### 3.1 NETWORK ARCHITECTURE

Based on the specialization of low-quality JPG-SR issue, our model can be separated into two functional stages: JPG recovering and SR generating. Hence, our SISR formulation consists in two stages:

$$I_{LR} = \mathcal{G}(I_{C-JPG}), I_{SR} = \mathcal{F}(I_{LR}). \tag{1}$$

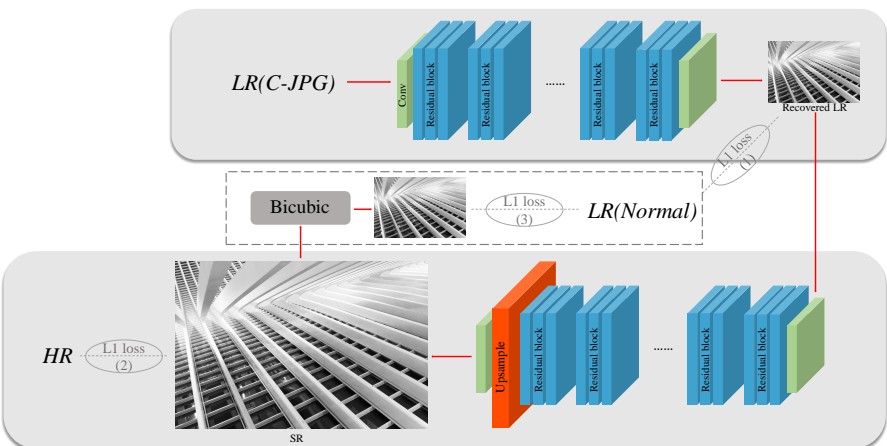

Figure 2: The illustration of the proposed JPG-SR model. The total model can be divided into the JPG recovering part, the SR generating part, and the reconstruction loss part.

As shown in Figure 2, training data are: C-JPG LR images ($LR(C - JPG)$); normal LR images ($LR(normal)$), which provide supervised information in the first stage; and HR images, which are the supervised information in the second stage.

**Stage I: JPG Recovering.** For low quality JPG images, the compression operation discards most of useful details. As a result, to rebuild details according to the original LR image plays a key role in this stage. To this end, we design a specialized network to learn the mapping between $LR(C-JPG)$ and $LR(normal)$.

The comparison of $LR(C - JPG)$ and $LR(normal)$ is shown in Figure 3. Given the original LR image, we firstly re-save it to its low quality JPG version as our training material. In detail, the JPG compression abandon information in Cb/Cr channel on every $8 \times 8$ patch. As shown in Figure 3, the size of the original LR image is 82K bytes, while its C-JPG version (80 percent lower in quality)

LR (Urban100)

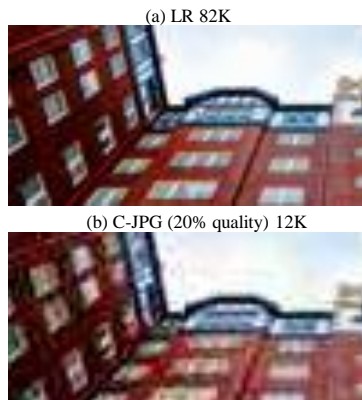

Figure 3: The comparison of compressed JPG image and its LR counterpart. We choose a typical building image from Urban100 dataset (Huang et al., 2015). Given the LR image, we leverage Pillow to compress it with 80 percent lower in quality. Benefiting from the compression process, C-JPG takes only 12K storage space, compared to the 82K LR image. However, the visual feeling of C-JPG displays more unpleasant details than LR.

is only 12K bytes. However, the quality of image are greatly decreased. This phenomenon can be clearly inferred from the visual contrast in Figure 3 that mussy details are generated in C-JPG image, such as the irregular shape of windows and the blurry object edge. Normally, to achieve better balance between storage cost and customer satisfaction, many websites provide images with partial-quality JPG ones. When performing SR operation on these images, the effect of these deuterogenic noises is enlarged, leading to aggressively unpleasant feeling in visual.

Since this recovering task shares similar goal with SR task, that is, trying to restore accurate information in pixel level, we employ an effective SR model, s-LWSR (Li et al., 2019), to address the details recovering issue. Learning from stage I in Figure 2, LR(C-JPG) images firstly go through the recovering model. In detail, a simple convolutional layer transfers the basic RGB layers into fixed numbers. The main body part of recovering stage shares the same structure with our SR generating: stacking 26 residual blocks with additional information pool to intensively extract information. Experiments in s-LWSR have proven the powerful learning ability of this architecture. Finally, a convolution layer inversely realize the transformation between middle layers and final 3 RGB layers.

Eventually, the recovered LR image for C-JPG achieves visual satisfactory in certain degree, and the artificial trace appears smooth, which greatly reduces the pixel inconsistency. Furthermore, SR generation will benefit from these better quality LR input deriving from C-JPG in stage I.

**Stage II: SR Generating.** Because SR generating acts as a complementary task and is not the main contribution in our paper, we just bring in a state-of-the-art SR method called s-LWSR (Li et al., 2019) in our model.

As is mentioned, s-LWSR leverages the combination of multi-level information from the front half of model. More details are shown in Figure 4. Specifically, chosen layers are stacked to form the information pool, which transfers the combination of abundant low-level information to high-level layers through a series of activation operations.

Thanks to stage I, the intermediate results partly deblur artificial noise brought by JPG compression process. During SR stage, intermediate images go through a similar model as recovering stage. However, in the last part, a specified upsampling block is applied to scale up the image to the ideal one. Normally, $2\times$, $3\times$, $4\times$, and $8\times$ are the most frequent SR tasks. Some models, such as MDSR (Lim et al., 2017), RCAN (Zhang et al., 2018b), can deal with all these scale-up tasks upon only one block with sharing parameters. In this paper, following most SR algorithms, we deal with single scale-up factor $4\times$ SR problem. As a result, the upsampling stage in our model contains two sub-pixel interpolation layers as in ESPCN (Shi et al., 2016). In detail, each upsample layer performs one $2\times$ scale-up. Finally, we obtain SR generations with satisfactory visual quality.

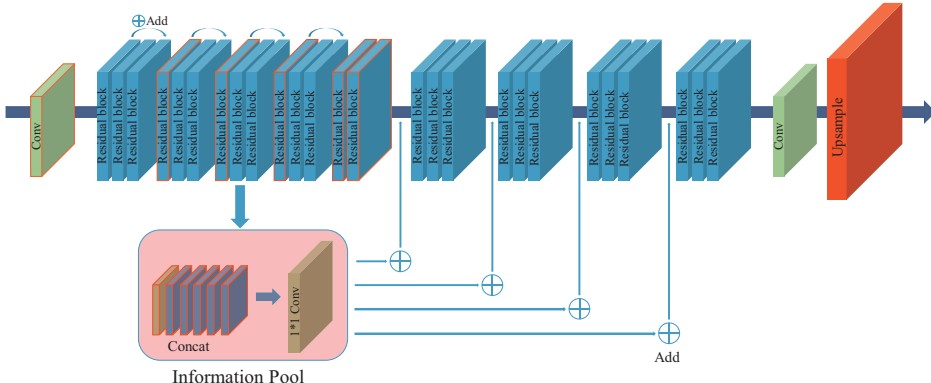

Figure 4: The s-LWSR architecture (Li et al., 2019).

## 3.2 LOSS FUNCTIONS

The framework of our approach includes two typical tasks: JPG recovering and SR generating, involving three similarity loss functions as follows.

In stage I, each pixel in C-JPG image corresponds to one counterpart in non-JPG LR image. The aim is to recover accurate value in pixels. As a result, we choose the following L1 loss:

$$\mathcal{L}_{L1}^1(I^{C-JPG}, I^{LR}) = \frac{1}{WH} \sum_{w=1}^{W} \sum_{h=1}^{H} \left\| I_{wh}^{LR} - G_{Rec}(I_{wh}^{C-JPG}) \right\|, \tag{2}$$

where $W$ and $H$ refer to the width and height of image respectively. $G_{Rec}$ is the transformation of stage I.

On the basis of pre-processed C-JPG, we further scale the intermediate ($LR_{rec}$) up to the default size of HR image through SR operation. Since our SR model pursues a better accurate SR generation, loss function in stage II inherits from that of former outstanding SR models:

$$\mathcal{L}_{L1}^2(I^{LR_{rec}}, I^{HR}) = \frac{1}{s^2 WH} \sum_{w=1}^{sW} \sum_{h=1}^{sH} \left\| I_{sw \times sh}^{HR} - G_{SR}(I_{wh}^{LR_{rec}}) \right\| \tag{3}$$

where $s$ is the scale factor.

Different from normal SR task, C-JPG SR involves more difference between the input C-JPG and its final supervised HR image. Inspired by marvelous development of unsupervised style-to-style learning, like CycleGAN (Kim et al., 2017), CinCGAN (Yuan et al., 2018), and WESPE (Ignatov et al., 2018), we apply cycle loss as the third component in our final loss function. Taking CycleGAN as an example, images in different domains are transferred with inconsistent contents. Cycle loss can keep same content of the original image, while changing the style features, such as colors and texture. In this paper, SR generations are downscaled to input size to further compare with the corresponding non-JPG LR images. Here, we also use L1 loss, and $Bic$ refers to bicubic interpolation downsampling.

$$\mathcal{L}_{L1}^3(I^{SR}, I^{LR}) = \frac{1}{s^2 WH} \sum_{w=1}^{sW} \sum_{h=1}^{sH} \left\| I_{wh}^{LR} - Bic(I_{sw \times sh}^{SR}) \right\|. \tag{4}$$

The final loss function is the combination of three L1 losses above with equal weight:

$$L_{total} = \mathcal{L}_{L1}^1(I^{C-JPG}, I^{LR}) + \mathcal{L}_{L1}^2(I^{LR_{rec}}, I^{HR}) + \mathcal{L}_{L1}^3(I^{SR}, I^{LR}), \tag{5}$$

## 4 EXPERIMENTS

In this section, we first describe the implementation details of our model. Then, we further analyze effects of all the proposed strategies in our model by ablation study. Finally, we compare our method with other leading SR models to validate the advantage.

### 4.1 IMPLEMENTATION DETAILS

**Dataset and Pre-processing Models.** As mentioned in the modeling part, the training process is based on compressed LR (C-JPG), LR, and HR images. Following other leading SR methods, we choose DIV2K Agustsson & Timofte (2017) as our training dataset. There are 800 HR-LR training image pairs and 100 validation images. To get the C-JPG LR images, all LR images are firstly processed by pillow package in python. In order to exhibit more contrast results with different compression levels, we respectively save JPG images with $20\%$ and $50\%$. Moreover, we pre-process C-JPG with MatLab2018, which contains the default deblurring model. All images (C-JPG LR and LR) are processed by data augmentation strategy which randomly rotates images by $90°$, $180°$, $270°$ and flips horizontally.

**Training Settings.** During training, images in C-JPG and LR dataset are randomly cropped with size of $48 \times 48$, while crop HR images with size of $192 \times 192$. Considering the computation and time consumption, we set batch size as 16. Testing experiments are done on most popular SR datasets: Set5 Bevilacqua et al. (2012), Set14 Zeyde et al. (2010), BSD100 Martin et al. (2001), and Urban100 Huang et al. (2015). Moreover, in order to prove the practical effectiveness, we perform our approach on the downloaded JPG images from the Internet. For both of the two stages, we apply Adam optimizer (Kingma & Ba, 2014), where $\beta_1 = 0.9$ and $\beta_2 = 0.999$. The learning rate is initialized as $1 \times 10^{-4}$ and half decreased every $2 \times 10^2$ iterations of back-propagation. Our model is trained over $1 \times 10^3$ times until reaching its convergence. We implement our model by Pytorch with a Titan 2080Ti GPU.

**Evaluation Standards.** For accurate related SR models, the most common evaluation standards are PSNR and SSIM Wang et al. (2004). Aligning with these algorithms, we evaluate our SR results on Y channel (i.e., luminance) of transformed YCbCr space instead of direct RGB channels.

### 4.2 ABLATION STUDY

To clarify the effectiveness of the recovering stage in our model, we apply ablation study to our model. Two strategies are employed. First, the recovering stage is removed from the model. Second, we use denoising with Matlab2018 to replace the recovering stage.

#### 4.2.1 STRAIGHT TRAINING

The recovering stage plays the crucial role in our model. In this part, we remove it from the model. As a result, C-JPG inputs directly go through the stage II: SR generating. Correspondingly, we just train the model with $\mathcal{L}_{L1}^2$ loss. The remaining architecture is the same. As shown in Fig.5, there are a lot of undesired artifacts compared with C-JPG model. Both of the PSNR and the SSIM decrease a large number. In fact, single model is hard to solve SR and recovering the C-JPG image simultaneously. The huge difference in supervised information leads to large variance among middle layers, which represent ideal details serving the final SR model. More comparisons are shown in the Appendix.

#### 4.2.2 PRE-DENOISING

To further analyze the learning ability of our recovering stage, we replace it by the denoising code in Matlab2018. The C-JPG images are firstly processed to a set of clean intermediate inputs: $I^{C-JPG} \rightarrow I^{inter}$. Then, we train the SR model with $I^{inter}$ and $I^{HR}$ pairs. The corresponding results are shown in Fig.5. It clearly illustrates that pre-denoisng operation can remove many artifacts, but it is still obviously worse than our model. In our opinion, more details should be recovered instead of just removing noise. Denoising only makes these C-JPG inputs clearer, while recovering brings accurate information to C-JPG images.

Table 1: $4\times$ scaled performance comparison on the validation datasets among different methods (PSNR/SSIM). Scores in bold indicate the best performance. M refers to pre-processed inputs, and C-JPG means that the model is trained on 20% quality compressed JPG dataset.

| Methods | Set5 | Set14 | BSD100 | Urban100 |
|---|---|---|---|---|
| DBPN | 24.63/0.6437 | 23.33/0.5629 | 23.79/0.5540 | 21.26/0.5372 |
| DBPN(M) | 25.80/0.7135 | 24.17/0.6060 | 24.42/0.5735 | 22.08/0.5845 |
| RCAN | 24.45/0.6355 | 23.17/0.5559 | 23.65/0.5367 | 21.12/0.5302 |
| RCAN(M) | 25.76/0.7123 | 24.15/0.6051 | 24.40/0.5725 | 22.04/0.5836 |
| s-LWSR$_{32}$ | 24.73/0.6487 | 23.44/0.5669 | 23.86/0.5467 | 21.39/0.5424 |
| s-LWSR$_{32}$(M) | 25.82/0.7153 | 24.21/0.6087 | 24.43/0.5742 | 22.10/0.5860 |
| s-LWSR$_{32}$(C-JPG) | 26.34/0.7488 | 24.58/0.6275 | 24.63/0.5874 | 22.54/0.6186 |
| Ours | **26.37/0.7525** | **24.61/0.6286** | **24.64/0.5882** | **22.55/0.6208** |

### 4.3 COMPARISON WITH LEADING METHODS

We provides quantitative evaluation of our model on four public benchmark datasets: Set5 (Bevilacqua et al., 2012), Set14 (Zeyde et al., 2010), BSD100 (Martin et al., 2001), and Urban100 (Huang et al., 2015). Because our research is SR generation from compressed JPG image, we firstly process images of the mentioned datasets to 20% quality JPG ones. Then, we compare our model with the sate-of-the-art SR methods: EDSR (Lim et al., 2017), DBPN (Haris et al., 2018), and RCAN (Zhang et al., 2018b). We present all comparison results in Figure 5 and Table 1. The new proposed method surpasses all the former methods in a large margin. More slippy details and structured shapes exist in our SR generations. Our model successfully recovers detailed features and edges of $I^{HR}$ and exhibits satisfactory results compared with former SR models. More results are shown in the Appendix.

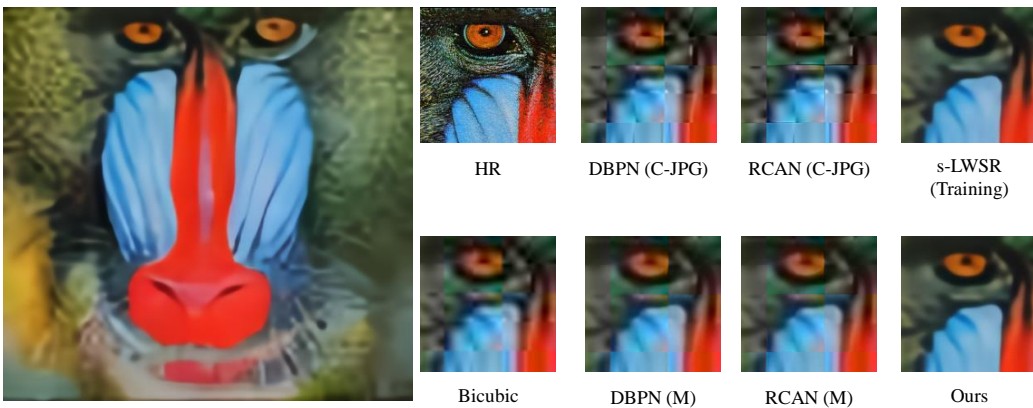

Figure 5: The comparison of C-JPG SR model and other SR methods. Inputs are divided into 20% quality JPG images (C-JPG) and Matlab pre-denosing images (M). Details show that our model achieves the best performance compared with others.

### 5 CONCLUSION

In this paper, we propose a lossless SISR model for low-quality C-JPG images which is extensive used on the Internet. Based on our redefined C-JPG SR pipeline, two functional stages are integrated to fulfill the SR task on C-JPG images. In addition, we employ cycle loss to guarantee the consistency after above two stages. The intensive experiments demonstrate that our model can learn capable representations of LR inputs for C-JPG SR task and outperform other cutting edges in SISR. More exploration should be executed on other CV tasks with C-JPG images inputs as the future work.

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

# A APPENDIX

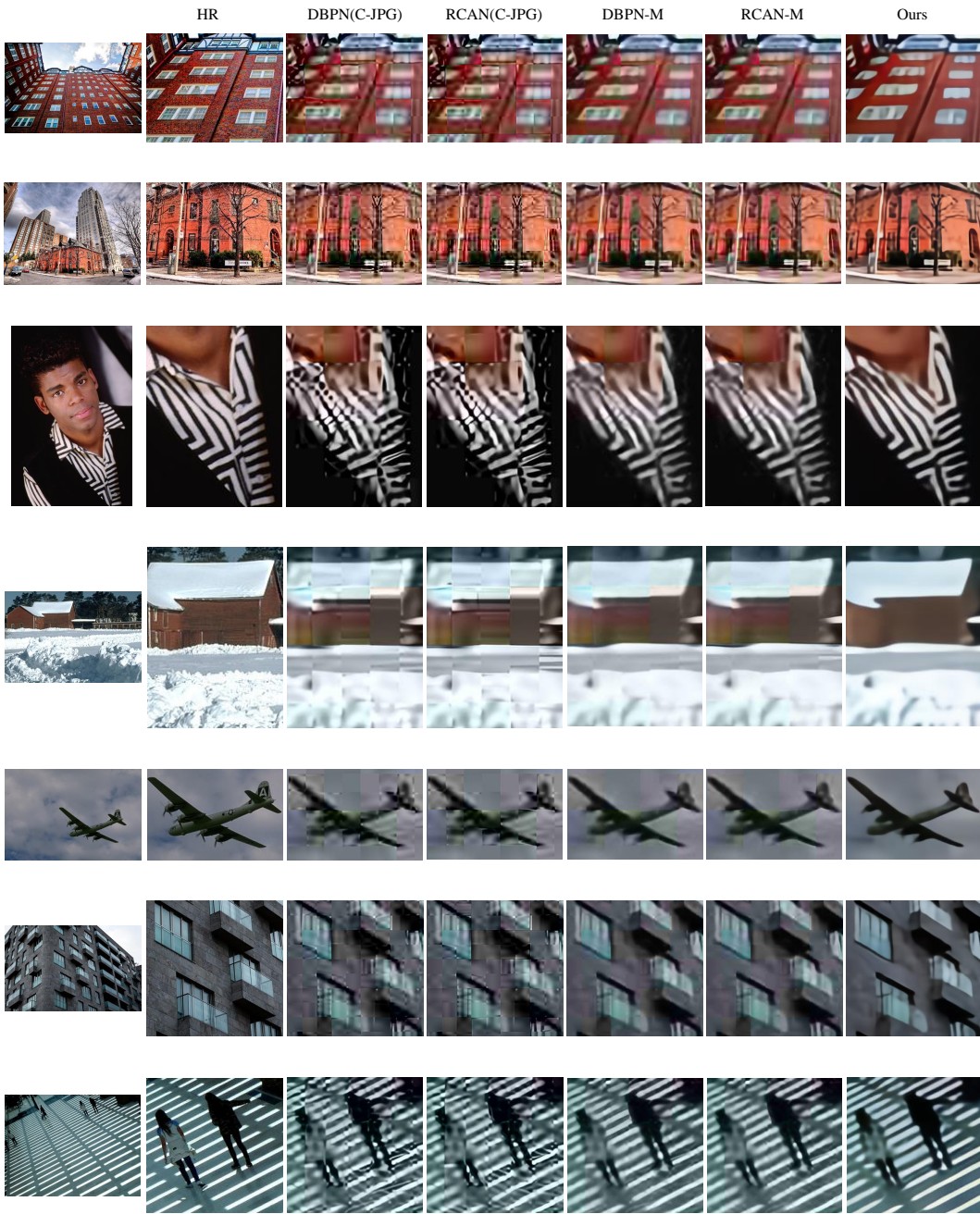

Figure 6: More comparisons of our C-JPG SR model and other leading SR methods. Inputs chosen from BSD100 Martin et al. (2001) and Urban100 Huang et al. (2015) are divided into 20% quality JPG images (C-JPG) and Matlab pre-denosing images (M).

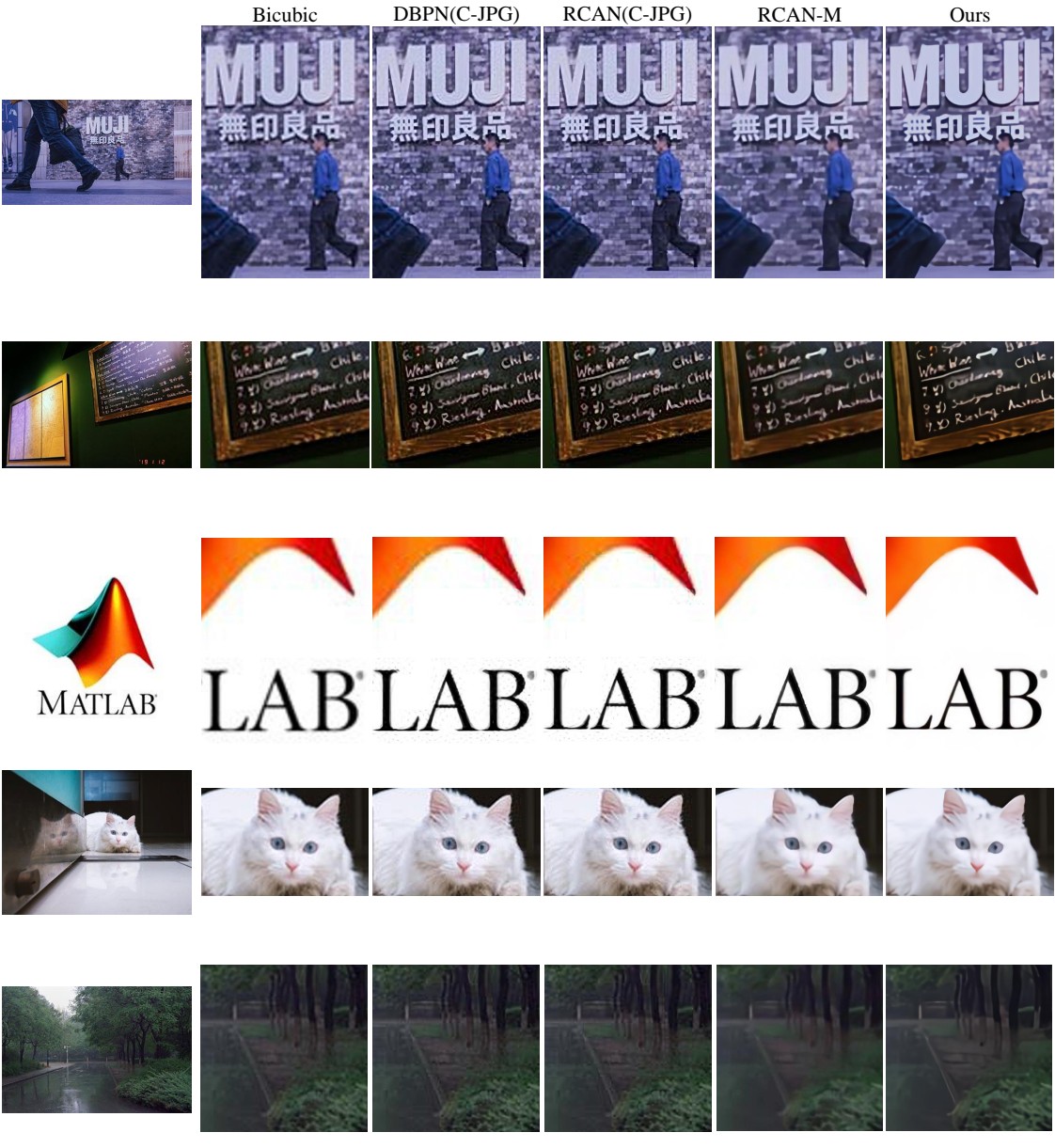

Figure 7: More comparisons of our C-JPG SR model and other leading SR methods. Inputs chosen from the Internet are divided into 20% quality JPG images (C-JPG) and Matlab pre-denosing images (M). It can be seen that our model achieves the best performance compared with others.

