# OpenReview forum: "LOSSLESS SINGLE IMAGE SUPER RESOLUTION FROM LOW-QUALITY JPG IMAGES"
_ICLR.cc/2020/Conference — Reject_

### Official Review · AnonReviewer2 · 2019-10-23
**Official Blind Review #2**

**Rating:** 3

**Review:**

This paper addresses the task of single image super resolution (SISR) on compressed JPG images. Different from the standard SISR problem, the input images of this task are compressed according to the JPEG standard, which have lower quality than what standard SISR deals with. The authors proposed a two-stage network which recovers the lossed information of compressed JPG (C-JPG) on the first stage and handles standard SR on the second stage. The whole model is trained with three L1 losses that ensure compressed information recovering, super resolution and LR-SR cycle consistency, respectively. Experiments on standard benchmarks demonstrate the effectiveness of this work.

However, my main concerns on this work are: 1. this works is not technically sound w.r.t. its novelty; 2. the efficacy of each loss function is not well supported by ablation studies and 3. the comparison experiments with other methods are not clearly stated.

**Main arguments:**

1. This works lacks novelty since the main idea is just a simple concatenation of a JPEG-artifact-removal model and a super resolution model. Both of the components are based on former work (s-LWSR 32). There’re no new insights on model design or inter-task relationship analysis provided. Besides, there’re already mature solutions on JPEG-artifact-removal and super resolution, like [R1-4]. Given the fact that cascading these solutions could also solve the C-JPG super resolution problem, the practical usage of this work is limited.
2. The efficacy of each loss function is not well supported. For example, what if we remove the $\mathcal{L}^1_{L1}$ loss defined in equation (2)? The performance could still be better than the other counterparts since the two-stage model have two times more parameters than a single super resolution net. Likewise, the effectiveness of the cycle consistency loss $\mathcal{L}^3_{L1}$ is also unclear.
3. In section 4.3, the input of other leading methods during training and testing is not described clearly. For fair comparison, it should be the same as `ours`. Otherwise, the whole comparison is invalid.

[R1] RESIDUAL NON-LOCAL ATTENTION NETWORKS FOR IMAGE RESTORATION, ICLR 2019.
[R2] Residual Dense Network for Image Restoration, arXiv 2018.
[R3] MemNet: A Persistent Memory Network for Image Restoration, ICCV 2017.
[R4] Beyond a Gaussian Denoiser: Residual Learning of Deep CNN for Image Denoising, TIP 2018.

**Experience Assessment:**

I have published in this field for several years.

**Review Assessment: Checking Correctness Of Derivations And Theory:**

N/A

**Review Assessment: Checking Correctness Of Experiments:**

I carefully checked the experiments.

**Review Assessment: Thoroughness In Paper Reading:**

I read the paper thoroughly.

---

### Official Review · AnonReviewer1 · 2019-10-24
**Official Blind Review #1**

**Rating:** 6

**Review:**

This paper focuses on the super-resolution (SR) task: to get better super-resolution images from compressed JPG inputs. They propose a two-stage pipeline to first recover the details that were lost during the compression of low-quality JPG images (called as JPG recovering), second to generate SR with satisfactory visual quality (called as SR generating). Finally, they added a hybrid loss function to support an integrated model and the results of their experiments showed good SR generation.

My decision is weak accept, considering the below aspects.
Positive points: (1) Recovering model methodology is quite novel compared to traditional noise- elimination-based approaches. (2) Experimental results showed better results compared to state-of-the-art methods. Cycle loss seems to be working well. Also, visualizations in the appendix showed better SR generation. (3) The paper is well organized.

For the experiments, the following should be addressed:
1.	Could you explain what does it mean, “Our model is trained over 1 × 10^3 times until reaching its convergence in training settings.”
2.	“In order to remove ring, checkerboard effects, as well as other noise, the former half sub-model is trained with pre-processed C-JPG LR images as inputs.” May I know preprocessing is employed here?
3.	Can you provide further explanations to statements in 4.2.1, 4.2.2? - “The huge difference in supervised information leads to large variance among middle layers, which represent ideal details serving the final SR model.”, “Denoising only makes these C-JPG inputs clearer, while recovering brings accurate information to C-JPG images.” are there any empirical results to support these statements”


**Experience Assessment:**

I do not know much about this area.

**Review Assessment: Checking Correctness Of Derivations And Theory:**

I assessed the sensibility of the derivations and theory.

**Review Assessment: Checking Correctness Of Experiments:**

I did not assess the experiments.

**Review Assessment: Thoroughness In Paper Reading:**

I made a quick assessment of this paper.

---

### Official Review · AnonReviewer3 · 2019-11-05
**Official Blind Review #3**

**Rating:** 1

**Review:**

In this article, the authors propose a single image super-resolution network that can generate high-resolution images from the corresponding C-JPG images. The method contains two main parts, namely a JPG recovering step, which recovers the information from low-quality JPG images, and an SR generation step, which generates SR images from the images achieved by the recovering step. Moreover, the authors leverage a cycle loss to generate better results.
The main contribution of this work is only the integration of existing models. The authors claimed that the proposed method is lossless, while there is no evidence to demonstrate it. The authors should show more evidence about the JPG recovering step, like how much information it can recover. Moreover, the SR generation step only incorporates s-LWSR without any improvement. It makes SR in this manuscript more like an application for the JPG recovering method rather than a contribution to the SISR field.
In the experimental section, Figure 5 makes readers confused. Does the image entitled “s-LWSR(Training)” mean the “STRAIGHT TRAINING” described in section 4.2.1? If yes, is there any perceptual difference between the result of s-LWSR and the result of ours? It is suggested that the authors should reorganize the results and provide more instructions. In Table 1, the results derived from s-LWSR32(C-JPG) and the proposed are very similar. The authors should more convincingly show the advantages of the proposed method. From the results, I observe that the SR images of the proposed model are blurry and lack much information about textures. At the same time, there are some other SISR studies, especially GAN based models like ESR-GAN, which show visual quality with more realistic and natural textures. I hope the authors can conduct more comparisons with these methods.
There are still some issues as follows:
1.	The authors should carefully check the format of the references in the whole article. For instance, in section 4.1, almost all references are in the wrong format. The same mistake happens in the caption of Figure 6. Please check the full article before submission.
2.	(Page 1, line 2 from bottom) Please add a reference to “bicubic”.
3.	(Section 3, line 1) Please add a full stop after “Challenge Formulation”.
4.	(Figure 2) Please enlarge the arrow of the red lines. They are hard to read right now.
5.	(Figure 4) The figure seems to miss the skip connect of the former five layers, which should be a part of the input added to the latter four layers (Li et al., 2019). In addition, please enlarge the arrow of the blue lines.
6.	(Section 4.2.1, line 4) “Both of the PSNR …” Does figure 5 can reflect this? Please add data instruction.


**Experience Assessment:**

I have read many papers in this area.

**Review Assessment: Checking Correctness Of Derivations And Theory:**

I carefully checked the derivations and theory.

**Review Assessment: Checking Correctness Of Experiments:**

I assessed the sensibility of the experiments.

**Review Assessment: Thoroughness In Paper Reading:**

I read the paper thoroughly.

---

> ### Author Response · Authors · 2019-11-14
> **Rebuttal**
>
> Thank you for your kind remarks and suggestions. In the paper, we focus on a novel SR issue deriving from the practical SR application. None of similar work has been done on condition that there are existing JPG compression removal model and SR generating model. We have tried all existing SR models and can’t obtain satisfied result. To overcome this issue, we propose a novel architecture with two separated functional sections based on triple inputs (C-LR, LR, HR). A circle loss is introduced to contribute the model training.
> As referred in our paper, most images on the Internet are pre-compressed. If we print them on cloth material, the unpleasant details will obviously appear.
> To evaluate the improvement of our model, we display the final results on Table.1 by PSNR and SSIM which are widely used as the accurate measurement of images.
> In this paper, our research mainly focus on how to solve the proposed C-JPG SR task. In other words, our C-JPG SR architecture is suitable for all former SR models. When we encounter this barrier, we try to use the denoising function of MATLAB2018 which is also involved in our paper as the comparison method to overcome it.
> As to “s-LWSR(Training)” mean the “STRAIGHT TRAINING” described in section 4.2.1, it is referred in the first line of 4.2.1: “The recovering stage plays the crucial role in our model. In this part, we remove it from the model.” Maybe the name will confuse readers, we will find a better one to succinctly express.
> Referred to GAN-related SISR models, the related work section mentioned some relative models. As we know, GANs leads to more blurry unpleasant details, which is a common sense in most GAN-related CV models. As a result, although GAN-SR methods generate photo-realistic images, it is not suitable for printing on cloth. In our paper, we pursue more accurate SR generations as most former methods.
> There are some issues as referred in the suggestions, and we will modify these.

---

### Decision · Program_Chairs · 2019-12-19

**Decision:**

Reject

**Comment:**

Main summary:  Sngle image super-resolution network that can generate high-resolution images from the corresponding C-JPG images

Discussions
reviewer 3: reviewer has a few issues including, claim the method is lossless, want more information about JPG revovering step
reviewer 1: (not knowledgable): paper is well written and reviewer gives very few cons
reviewer 2: main concerns are wrt novelty and technically sound
Recommendation: the 2 more knowledgable reviwers mark this as Reject, I agree.